# Cyclic GMP-AMP Synthase (cGAS) Deletion Promotes Less Prominent Inflammatory Macrophages and Sepsis Severity in Catheter-Induced Infection and LPS Injection Models

**DOI:** 10.3390/ijms26115069

**Published:** 2025-05-24

**Authors:** Chatsuree Suksamai, Warerat Kaewduangduen, Pornpimol Phuengmaung, Kritsanawan Sae-Khow, Awirut Charoensappakit, Suwasin Udomkarnjananun, Sutada Lotinun, Patipark Kueanjinda, Asada Leelahavanichkul

**Affiliations:** 1Medical Microbiology, Interdisciplinary and International Program, Graduate School, Chulalongkorn University, Bangkok 10330, Thailand; s.chatsuree@gmail.com; 2Center of Excellence on Translational Research in Inflammation and Immunology (CETRII), Department of Microbiology, Faculty of Medicine, Chulalongkorn University, Bangkok 10330, Thailand; porpluemw@gmail.com (W.K.); pphuengmaung@gmail.com (P.P.); kritsanawan_29@hotmail.com (K.S.-K.); awirut.turk@gmail.com (A.C.); 3Division of Nephrology, Department of Medicine, Faculty of Medicine, Chulalongkorn University and King Chulalongkorn Memorial Hospital, Bangkok 10330, Thailand; suwasin.u@gmail.com; 4Center of Excellence in Skeletal Disorders and Enzyme Reaction Mechanism, Department of Physiology, Faculty of Dentistry, Chulalongkorn University, Bangkok 10330, Thailand; sutada.l@chula.ac.th; 5Department of Pathology, University of Massachusetts Medical School, Worcester, MA 01605, USA

**Keywords:** cGAS, macrophage, OXPHOS, LPS, sepsis model

## Abstract

Activation of cGAS, a cytosolic receptor recognizing double-stranded DNA, in macrophages is important in sepsis (a life-threatening condition caused by infection). The responses against sepsis induced by subcutaneous implantation of the *Pseudomonas*-contaminated catheters in cGAS-deficient (cGAS^−/−^) mice were lower than in wild-type (WT) mice as indicated by liver enzymes, white blood cell count, cytokines, and M1-polarized macrophages in the spleens. Likewise, a lethal dose of lipopolysaccharide (LPS) induced less severe sepsis severity as determined by mortality, organ injury, cell-free DNA, and serum cytokines. Patterns of the transcriptome of lipopolysaccharide (LPS)-stimulated bone marrow-derived macrophages were clearly different between cGAS^−/−^ and WT cells. Gene set enrichment analysis (GSEA; a computational statistical determination of the gene set) indicated more prominent enrichment of oxidative phosphorylation (OXPHOS; the mitochondrial function) and mTORC1 pathways in LPS-activated cGAS^−/−^ macrophages compared with WT. Meanwhile, LPS upregulated cGAS and increased cGAMP (a cGAS inducer) only in WT macrophages along with less severe inflammation in cGAS^−/−^ macrophages, as indicated by supernatant cytokines, pro-inflammatory molecules (nuclear factor kappa B; *NF-κB*), M1 polarization (*IL-1β*, CD80, and CD86), and macrophage extracellular traps (METs; web-like structures composed of DNA, histones, and other proteins) through the detection of citrullinated histone 3 (CitH3) in supernatant and immunofluorescent visualization. In conclusion, less prominent pro-inflammatory responses of cGAS^−/−^ macrophages than WT were demonstrated in mice (catheter-induced sepsis and LPS injection model) and in vitro (transcriptomic analysis, macrophage polarization, and METs).

## 1. Introduction

Sepsis, an extreme response to infection regardless of the type of microorganism, is a life-threatening condition with multi-organ failure and high mortality [1]. While macrophages play a crucial role in innate immune defense through several pattern recognition receptors (PRRs), macrophage hyperimmune responses worsen sepsis, and the modulation of macrophages attenuates sepsis through various molecular mechanisms [2,3,4]. With a simplified macrophage polarization classification (M1 pro-inflammatory and M2 anti-inflammatory polarizations), controlling macrophage responses through different molecular pathways to direct them toward a proper response is one of the interesting strategies to attenuate inflammatory diseases [5,6]. Cyclic guanosine monophosphate–adenosine monophosphate (GMP-AMP) synthase (cGAS), one of the PRRs in several cells, is a cytosolic sensor that recognizes fragmented DNAs, especially the microbial DNA, and initiates the synthesis of cyclic GMP-AMP (cGAMP), a secondary messenger that activates stimulator of interferon genes (STING) [7,8,9]. In addition to recognizing pathogen-associated molecular patterns (PAMPs), cGAS detects damage-associated molecular patterns (DAMPs) that are released from stressed or dying cells, such as apoptotic cells, resulting in inflammatory cytokine production [10,11]. While cGAS recognizes intracellular organisms (bacteria and viruses), cGAS also detects mitochondrial DNA (mtDNA) and damaged host DNA (leaked from the nuclei) as DAMPs [12,13]. Interestingly, overwhelming inflammatory stimuli can drastically alter cell energy status and possibly cause mitochondrial damage, releasing mtDNA that could escalate the responses, similar to a positive feedback cycle leading to more severe inflammation [14,15]. Hence, interference with the cGAS–STING pathway attenuates inflammation in several conditions, either non-infectious causes or infectious etiologies [16]. For example, several uremic toxins, which accumulate in the blood following the removal of both kidneys, induce macrophage responses with systemic inflammation, partly through the activation of cGAS [13]. In cecal ligation and puncture (CLP)-induced sepsis, cGAS recognized mtDNA in stressed lung epithelial cells, thereby promoting lung inflammation via the STING pathway [12,17]. Hence, cGAS^−/−^ immune cells (neutrophils and macrophages) exhibit less prominent responses against several activators [13,18,19]. The blockage of the cGAS pathway might be beneficial in reducing inflammatory damage in several conditions.

Despite some similarities among sepsis from several causes, differences in pathogenesis and molecular pathways are possible. For example, peritonitis is closely associated with the translocation of organisms from the gut into the liver (gut-liver axis) and Kupffer cell responses, while pneumonia might mainly rely on alveolar macrophages that have naturally adapted to defend against different types of organisms [20], unlike Kupffer cells [21]. Therefore, supportive data on the role of cGAS in sepsis using different models are essential for translating this topic to the clinical setting. Catheter-related infection is common in patients [22], and the subcutaneous catheter insertion model in mice can induce sepsis with highly virulent organisms [23]. Although the subcutaneous catheter insertion mouse model is different from the patient situation because the catheters are not placed inside the vessels, this model is used as a preclinical model to explore catheter-associated infection [24] due to the small diameter of mouse vessels [25]. Also, the catheter insertion model can apply to other clinical situations with subcutaneous foreign body insertion in humans; for example, the Tenckhoff catheter in patients with peritoneal dialysis [26] that may develop sepsis [27]. Thus, the catheter insertion model, a simpler intervention compared to surgical models of sepsis (e.g., the CLP model), is one of the interesting sepsis models. Because sepsis from the Gram-negative bacteria is usually more severe than Gram-positive bacteria [28], and due to our previous successful induction of sepsis using *P. aeruginosa*-infected catheters [29], *Pseudomonas*-infected catheters were subcutaneously implanted in WT and cGAS^−/−^ mice to represent sepsis from subcutaneous catheter infection. In addition, the sepsis model using a lethal dose of lipopolysaccharide injection, one of the commonly used mouse models of sepsis [30], was also conducted. Furthermore, in vitro and transcriptomic analyses of cGAS^−/−^ macrophages were also performed.

## 2. Results

### 2.1. Deletion of cGAS Reduces Disease Severity in a Catheter-Induced Sepsis Mouse Model

Because mitochondrial injury induces the release of mitochondrial DNA (as DAMPs) [31], which activates cGAS [13], we hypothesized that sepsis would be less severe in cGAS^−/−^ mice compared to wild-type (WT) mice. To test this hypothesis, a sepsis model using subcutaneous implantation of *Pseudomonas aeruginosa*-contaminated catheters into cGAS^−/−^ and WT mice for 5 days, mimicking catheter-associated infections observed in humans, was established (Figure 1A, left side). There was no skin inflammation at 5 days after the implantation in any mice compared with the post-catheter insertion time point (day 1) (Figure 1A, right side). However, the catheters from the infected catheter group demonstrated blue–green-colored pus (Figure 1A, right side in the red-colored frame), while the control catheter showed clearer coloring without pus (Figure 1A, right side in the blue-colored frame). The baseline levels of all parameters were similar between cGAS^−/−^ and WT, while sepsis induced higher levels of all parameters compared to the baseline, except for serum creatinine (Figure 1B), including liver enzymes (Figure 1C,D), cells in the complete blood count (CBC) (Figure 1E–G), bacteremia (Figure 1H), endotoxemia (Figure 1I), cell-free DNA (Figure 1J), and serum cytokines (TNF-α, IL-6, and IL-10) (Figure 1K–M). The inflammatory responses against sepsis were less severe in cGAS^−/−^ compared to WT mice, as indicated by reduced (i) liver dysfunction, using serum aspartate transaminase (AST) and alanine transaminase (ALT) levels, (ii) total white blood cells (WBC) and lymphocyte counts, and (iii) cytokine levels (TNF-α, IL-6, and IL-10). However, no significant differences were observed in renal function (serum creatinine) (Figure 1B), blood neutrophil abundance (Figure 1F), bacteremia (Figure 1H), endotoxemia (Figure 1I), and cell-free DNA (Figure 1J). Flow cytometry analysis of the spleens showed that baseline flow cytometric characteristics were similar between cGAS^−/−^ and WT mice. Using F4/80 as a macrophage biomarker, septic WT mice showed an elevation in M1 macrophages (CD80 and CD86 positive cells) (Figure 2A,B) and activated macrophages (IAb or MHC-class II positive cells) (Figure 2F), with no obvious change in M2 anti-inflammatory macrophage parameters (elevated CD163 but reduced CD206) (Figure 2C,D). Analysis with the CD11b marker, myeloid-derived macrophages (Gr-1 positive) (Figure 2G), inflammatory monocytes (Ly6c positive) (Figure 2H), and neutrophils (Ly6g positive) (Figure 2I) showed no significant differences between control and septic mice. To test sepsis severity, a lethal dose of LPS was injected. The 2-day mortality rate (Figure 3A) in LPS-injected WT mice was higher than in cGAS^−/−^ mice, while serum creatinine (Figure 3B) and ALT at 6 h post-LPS (Figure 3C) were not different. On the other hand, cell-free DNA (Figure 3D) and serum cytokines (TNF-α, IL-6, and IL-10) (Figure 3E–G) in LPS-injected WT mice were worse than in cGAS^−/−^ mice. These data indicated that sepsis-induced immune responses toward pro-inflammatory M1 macrophages were more prominent in the WT than cGAS^−/−^ mice, supporting the potential role of cGAS activation by mitochondrial DNA released during the overwhelming inflammatory responses in sepsis-induced hyperinflammation [32].

### 2.2. Transcriptomic Analysis Reveals Diminished IRF and JAK/STAT Activity, but Enriched OXPHOS and mTORC1 Pathways in cGAS-Deficient Macrophages

Given the observed reduction in disease severity in cGAS-deficient mice and the reported role of macrophages in catheter-associated infection [33], macrophages derived from bone marrow (BMDMs) from cGAS^−/−^ and WT mice were tested. The absence of cGAS expression in cGAS^−/−^ BMDMs was evidenced by lower cGAS gene expression and reduced levels of 2′3′-cGAMP (a secondary messenger synthesized by cGAS that activates the downstream STING pathway) (Figure 4A). With the 24 h stimulation by lipopolysaccharide (LPS; an important bacterial component), cGAS gene expression was upregulated and increased 2′3′-cGAMP levels only in WT macrophages, but not in cGAS^−/−^ cells (Figure 4A). These findings support the notion of cGAS activation by cytosolic DNA during sepsis [34]. Then, the RNA from macrophages was used for transcriptomic analysis to identify relevant molecular functions. As such, the transcriptomic pattern was obviously different between cGAS^−/−^ and WT cells, either with LPS activation or the control untreated (UT) conditions (Figure 4B,C). In the UT condition, more prominent gene expression was mostly demonstrated in the WT with the following lists: *Igkv* (immunoglobulin kappa variable cluster), *Tk* (thymidine kinase), *Kif* (kinesin superfamily proteins), *Top2a* (DNA topoisomerase II alpha), *Mki67* (marker of proliferation Ki-67), *Aunip* (aurora kinase A and ninein interacting protein), *Pimreg* (PICALM interacting mitotic regulator), Stab2 (a group of transmembrane receptor protein), and *Cks1b* (cyclin-dependent kinases regulatory subunit 1) (Figure 4B), indicating the possible differences in cell growth, proliferation, and deaths between cGAS^−/−^ and WT macrophages. With LPS stimulation, the more prominent gene expression was mostly demonstrated in the cGAS^−/−^ with the following lists: *Clmp* (CXADR-like membrane protein), *Marco* (macrophage receptor with collagenous structure), *Vnn* (vanin-1), *Pilrb* (paired immunoglobin-like type 2 receptor beta), *Draxin* (dorsal inhibitory axon guidance protein), *IL6* (interleukin 6), *Lad* (leukocyte adhesion deficiency), *IL12* (interleukin 12), *Shisa* (a transmembrane protein family), *Ppm* (metal-dependent Ser/Thr protein phosphatases), *Fscn* (fascin actin-bundling protein 1), *Mmp9* (matrix metalloproteinase-9), and *Tal2* (TAL BHLH transcription factor 2) (Figure 4B). Despite the differentially expressed gene analysis, the direction of macrophage function based on this list is still difficult to predict. However, the upregulated *Il6* and *Il12a* potentially elevated more prominent proinflammatory cytokines in LPS-treated cGAS^−/−^ macrophages than in LPS-treated WT (Figure 4B). Then, the bioinformatic tools, specifically DoRothEA [35] and PROGENy [36], were further used to estimate transcription factor and pathway activities, respectively, using the transcriptomic data. The results revealed diminished transcription factor activity of the intracellular antigenic recognition pathways, including the interferon regulatory factor (*IRF*) (e.g., *Irf1*, *Irf2*, and *Irf9*) and Janus kinase–signal transduction and activation of transcription (*JAK*/*STAT*) (e.g., *Stat1* and *Stat2*) in LPS-treated cGAS^−/−^ BMDMs compared to LPS-treated WT BMDMs (Figure 4C). Also, the activity of transcription factors associated with the synthesis of several proteins, including the nuclear factor kappa B (*NF-κB*) (e.g., *Rela* and *Nfkb1*) and mitogen-activated protein kinase (*MAPK*) (e.g., *Jun* and *Fos*), was also more prominently increased in cGAS^−/−^ macrophages (Figure 4C). Although *NF-κB* and *MAPK* are often mentioned in relation to pro-inflammation, both transcriptional factors can also play a role in anti-inflammatory processes [37,38].

Subsequently, the gene set enrichment analysis (GSEA) revealed enrichment of oxidative phosphorylation (OXPHOS) and mTORC1 pathways, indicating the significant up-regulation of genes related to both pathways in LPS-treated cGAS^−/−^ BMDMs compared to LPS-treated WT BMDMs (Figure 4D). Notably, OXPHOS is associated with the energy use of M2 macrophage polarization, and glycolysis is more closely related to M1 macrophage polarization [39,40], while the mTORC1 pathway is correlated with cell metabolism and protein synthesis, perhaps due to the M2 polarization. Hence, the absence of a functional DNA sensor in cGAS^−/−^ macrophages demonstrated some impacts not only on viral infection (IRF pathway) but also on LPS activation, possibly due to the presence of mitochondrial DNA in the cytosol after LPS induction [41].

### 2.3. cGAS Restricts Macrophages to M1-like Subtype

To further determine the function of macrophage polarization, the transcriptomic analysis using the MacSpectrum, a bioinformatic tool for predicting macrophage subtypes [42], was performed. Indeed, the LPS-treated cGAS^−/−^ BMDMs were positioned closer to an M2-like phenotype, in contrast to LPS-treated WT BMDMs that were closer to the M1-like phenotype (Figure 5A). Then, further in vitro experiments were performed to support the data from the transcriptomic analysis. As such, supernatant cytokines (TNF-α, IL-6, and IL-10) (Figure 5B,D), along with the expression of the *NF-κB* transcriptional factor of cGAS^−/−^ BMDMs, were lower than in WT (Figure 5E). Additionally, more prominent M1 macrophage polarization in WT BMDMs compared with cGAS^−/−^ cells was also indicated through the upregulated, *IL-1β* approximately 14-fold higher than LPS-activated cGAS^−/−^ cells (but not *iNOS*) using polymerase chain reaction (PCR) (Figure 6A,B) and the elevated CD80 (2.3-fold higher than cGAS^−/−^ cells) and CD86 (1.5-fold higher than cGAS^−/−^ cells) by flow cytometry analysis (Figure 6C,D). Among M2 macrophage polarization parameters, only *Arg-1* (Figure 6F), but not other parameters (*Fizz*, *TGF-β*, CD163, and CD206) (Figure 6E,G–J), was more prominently upregulated in LPS-stimulated cGAS^−/−^ BMDMs than in the WT. Notably, in comparison with the control, LPS-activated WT BMDMs demonstrated M1 polarization, as indicated by elevated *IL-1β*, *iNOS*, CD80, and CD86 (Figure 6A–D) without an alteration in M2 macrophage polarization parameters (Figure 6E–I). Because mitochondrial injury might be a source of DNA that stimulates cGAS in LPS-activated macrophages, as previously reported in the cGAS^−/−^ neutrophils [13], mitochondrial DNA (mtDNA) was measured. As such, supernatant mtDNA and fluorescent MitoTracker Red (the molecule that binds to the intact mitochondrial membrane) in LPS-treated BMDMs were lower than the control without the differences between cGAS^−/−^ and WT cells (Figure 7A,B). The extracellular flux analysis (the cell energy status) was also determined by oxygen consumption rate (OCR) (mitochondrial stress test) and extracellular acidification rate (ECAR) (glycolysis stress test) (Figure 7C,D). Indeed, there was a more prominent reduction and elevation of maximal respiration and glycolysis capacity, respectively, in LPS-activated WT macrophages compared with the cGAS^−/−^ cells (Figure 7E,F). These data indicated a similar LPS-induced mitochondrial injury between cGAS^−/−^ and WT cells; however, WT macrophages respond to the DNA more strongly than cGAS^−/−^ macrophages. In parallel, supernatant citrullinated histone 3 (CitH3) (Figure 7G) and macrophage extracellular traps (METosis) (Figure 7H) in LPS-activated WT BMDMs were more prominent than in cGAS^−/−^ cells (Figure 7I), supporting the more prominent inflammatory response of WT macrophages than the cGAS^−/−^ cells. Hence, LPS similarly damages mtDNA in both cGAS^−/−^ and WT cells, but the responses to mtDNA in WT cells were prominent enough to dominantly induce METosis. These findings supported that cGAS deletion promotes less predominant pro-inflammation in macrophages.

## 3. Discussion

Less prominent severity of sepsis in cGAS^−/−^ mice than in the WT was demonstrated in catheter-induced sepsis and LPS injection models, partly through the less severe macrophage responses, as supported by RNA sequencing analysis and other parameters.

### 3.1. Sepsis-Induced Mitochondrial Injury and the Less Prominent Catheter-Induced Sepsis and LPS Injection Models in cGAS^−/−^ Mice Compared with WT

Sepsis is an overwhelming response against microbes, regardless of the organismal causes [43], due to the similar responses against several PAMPs and DAMPs [44]. In catheter-induced sepsis, cGAS^−/−^ mice demonstrated lower serum cytokines and liver injury than WT with a similar burden of infection (bacteremia, endotoxemia, neutrophilia, and cell-free DNA). Notably, biofilms in the catheters are an important bacterial reservoir that can resist the washing step of the procedure (see method) and cause sepsis, as previously mentioned [23]. Meanwhile, cGAS^−/−^ mice demonstrated a better survival rate in the lethal LPS injection model. Hence, the lower severity of sepsis in cGAS^−/−^ mice than in WT might be due to the less prominent immune responses against the infection, especially the responses by macrophages, which are one of the major immune cells responsible for sepsis severity [45]. Although there might be potential off-target effects from other immune cell populations in cGAS^−/−^ mice different from the more myeloid cell-specific knockout model (e.g., LysM-Cre-driven cGAS deletion), macrophages are important immune cells responsible for sepsis [46], and studies in cGAS^−/−^ mice (an easier gene-manipulated model) can be used as proof-of-concept studies. As such, cGAS is one of the pattern recognition receptors that recognize DNA in the cytosol, which might be DAMPs, mitochondrial DNA (mtDNA), host DNA, or PAMPs (bacterial and viral DNA) (despite the prokaryotic DNA characteristics, mitochondria are organelles of eukaryotic cells) [47]. In sepsis, mitochondrial injury is possible through various conditions, including reactive nitrogen/oxygen species, overwhelming mitophagy, and impaired mitochondrial membrane permeabilization, while the exposure of mtDNA in the cytosol enhances inflammation by triggering several main signaling pathways, including toll-like receptors (TLR), NOD-like receptors, and cGAS [32]. Due to the presence of mtDNA in the cytosol, the activation of cGAS after mitochondrial damage is one of the important signaling pathways. Indeed, cGAS facilitates M1 pro-inflammatory macrophage polarization in various disease models, including myocarditis [48], lupus [19], and hepatitis B viral infection [49]. In sepsis, the M1 pro-inflammatory response is one of the underlying mechanisms inducing sepsis-induced systemic inflammatory response syndrome and septic shock, while the blockage of M1-polarized macrophages and the deviation of macrophage functions toward more M2 anti-inflammatory macrophages might be beneficial [6]. Although several pathways induce M1 macrophage polarization, cGAS is one of the interesting receptors that recognize the damage of DNA. Indeed, impacts of DNA damage in sepsis are increasingly mentioned as serum cell-free DNA is currently used as a severity and prognostic biomarker of sepsis [44]. However, serum cell-free DNA during sepsis is a combination of DNA from the host cells, mitochondria, and organisms (the translocation of bacterial DNA from the gut into the blood circulation is possible through the natural spontaneous DNA digestion into the smaller pieces of the DNA) [50]. Despite a small fraction of mtDNA in the serum cell-free DNA, the presence of mtDNA in the cytosol of macrophages during sepsis significantly induced systemic inflammation. Then, our data supported a possible beneficial effect of cGAS inhibitors [51] and/or the manipulation of macrophage polarization [5,52,53] as an adjuvant anti-inflammatory drug for Gram-negative bacterial sepsis. More studies are warranted.

### 3.2. Macrophage Anti-Inflammatory Direction in cGAS^−/−^ Compared with WT Cells

Because cGAS is the receptor for DNAs, particularly for host genomic DNA and mtDNA, which are among the important damage-associated molecular patterns (DAMPs) that alert innate immunity to the possible infection in the body [50]. In response to these DAMPs, macrophages (or monocytes) recognize these DNAs, which are translocated from the regular sites of the cell (nuclei and mitochondria), and transform from the naïve macrophages (M0) into pro-inflammatory macrophages (M1) [48]. The limitation of macrophages in transforming into pro-inflammatory cells in cGAS^−/−^ mice tips the balance of the immune responses toward more anti-inflammatory stages than the regular responses that might affect several conditions. From our in vitro results, the anti-inflammatory direction of cGAS^−/−^ macrophages against LPS was indicated by the reduction in several parameters, including supernatant cytokines, *NF-κB* transcriptional factor, M1 polarization (down-regulated *IL-1β* and decreased CD80 and CD86), and METosis, with an elevation in *Arg-1* (an enzyme involved in M2 anti-inflammatory macrophage polarization). The transcriptomic analysis demonstrated an obvious difference between cGAS^−/−^ and WT macrophages in either the resting state or after LPS activation (Figure 4B,C). In the untreated resting state, the WT macrophages demonstrated higher expression of several genes encoding the enzymes for cell proliferation (thymidine kinase and aurora kinase A) [54,55] and DNA replication (DNA topoisomerase) [3,56], indicating the possible limitation on cell proliferation of cGAS^−/−^ macrophages, at least in some conditions. With LPS stimulation, cGAS^−/−^ macrophages showed several upregulated genes compared with WT, including the pro-inflammatory genes (*IL6* and *IL12*), anti-inflammatory genes (*Lad* and *Ppm*) [57,58], and the genes with uncertain pro- or anti-inflammation (*Marco*, *Mmp9*, and *Vnn*) [59,60,61]. Although LPS more prominently upregulated the *Il6* and *NF-κB* genes in cGAS^−/−^ macrophages than in WT, the elevation of several molecules, such as *β-catenin* (a suppressor of ARID5A, a protein that stabilizes *Il6* mRNA) [62,63], might explain the discrepancy between upregulated *Il6* mRNA and the lower secreted IL-6 at the protein level. Indeed, the discrepancy between mRNA expression and protein level might be due to the different stability between mRNA and proteins (the shorter expression of mRNA with the more stable level of proteins) due to the discrepancy in various regulatory processes between RNA transcription and protein production [64,65]. Further studies with mRNA stability assays and/or protein degradation analyses [66,67] might be beneficial. Also, despite the generally pro-inflammatory roles of *NF-κB* transcriptional factors [68], anti-inflammatory effects from *NF-κB* might occur only in a few specific conditions [37,38].

Despite the discrepancy, the estimated expression of transcription factor genes using bioinformatic tools also indicates slight differences between cGAS^−/−^ and WT macrophages in the untreated control conditions, and the differences were more prominent after LPS stimulation. Because the transcriptional factors are used for both pro- and anti-inflammation, it is difficult to predict the direction of macrophage responses (pro- versus anti-inflammation) through these transcriptional factors. However, the lower expression of *Irf* and *Stat* genes in cGAS^−/−^ than in WT macrophages might partly correlate with the less inflammatory responses of cGAS^−/−^ cells against LPS. Then, further analysis was used. Indeed, with the waterfall plots from the GSEA, OXPHOS and mTORC1 pathways are enriched in LPS-treated cGAS^−/−^ BMDMs (Figure 4D). An increase in OXPHOS after LPS activation has previously been mentioned in macrophages [40] and natural killer (NK) cells [69], while an enrichment of mTORC1 might be associated with protein synthesis during the macrophage polarization [70] and metabolic reprogramming [71]. The inhibition of mTORC1 reduced OXPHOS and cytokine production in NK cells [72,73], and mTORC1 enhances glycolysis and subsequently induces mitochondrial biogenesis [74].

In support of a broader metabolic link between innate immune sensing and immune modulation, recent evidence demonstrates that STING activation enhances the pentose phosphate pathway (PPP) by stabilizing transketolase (TKT), thereby amplifying antiviral immune responses through metabolite-mediated signaling cascades [75]. While our data do not directly interrogate PPP involvement in bacterial sepsis, this finding aligns with our observation of increased OXPHOS activity and suggests that similar metabolic shifts may shape inflammatory polarization in macrophages through the cGAS-STING axis. In M2 macrophage polarization, glycolysis is not strictly required as long as the OXPHOS pathway remains intact [76]. Thus, both OXPHOS and mTORC1 enhancement here possibly supported the more prominent anti-inflammatory macrophages of cGAS^−/−^ BMDMs compared with WT. Notably, LPS did not induce M2 polarization in cGAS^−/−^ BMDMs but rather deviated the responses toward more anti-inflammatory characteristics compared with LPS-stimulated WT cells, as indicated by the incomplete elevation of M2 parameters (only on the MacSpectrum output and elevated *Arg-1* but not by other factors). As such, the mTORC1 enrichment may prime LPS-treated cGAS^−/−^ BMDMs for the OXPHOS-mediated cell activities with the synthesis of several proteins [77]. The more prominent OXPHOS in LPS-induced cGAS^−/−^ macrophages over WT cells was also supported by the extracellular flux analysis (Figure 7E). It is interesting to note that previous studies have shown that cGAS activation promotes mTORC1 signaling and M1-like responses [70]. In the current study, the deletion of cGAS in macrophages also elevated mTORC1 more than the WT macrophages with intact cGAS receptors after LPS stimulation. However, these cGAS-deficient macrophages exhibited less prominent M1-polarized characteristics, as demonstrated by upregulated *Arg-1*, an energy shift away from the glycolytic phenotype (Appendix A), and an increased OCR/ECAR ratio (Appendix A) compared to the WT cells. In addition to mTORC1, Akt is also important in M2 polarization. Overexpression of Akt in constitutively active mTORC1 BMDMs induces *Arg-1*, *Fizz-1*, *Mgl2*, and *Mgl1* [77]. Moreover, attenuation of STING augments Akt signaling activation, as demonstrated in breast cancer cells [78]. Taken together, these studies suggest that cGAS deletion may promote Akt activation and induce M2 polarization despite mTORC1 activation. Future studies in macrophages are required.

Nevertheless, more severe impairment of mitochondrial function (as shown by extracellular flux analysis) in LPS-treated cGAS^−/−^ macrophages compared with WT was previously demonstrated [12]. Although the damage to mtDNA after LPS activation in cGAS^−/−^ macrophages and WT was not different, as indicated by mtDNA expression and the MitoTracker Red assay (Figure 7A,B), the responses to these mtDNA in cGAS^−/−^ macrophages were lower than the WT cells, as demonstrated by the less prominent M1 macrophage polarization parameters (Figure 6). Because of the importance of cell energy toward macrophage polarization and other pro-inflammatory activities [79], METs (another interesting pro-inflammatory function) were explored. The lower expression of *Irf* and *Stat* in cGAS^−/−^ than WT macrophages might be partly responsible for the lower METs in cGAS^−/−^ cells, as both genes can induce METs with several downstream activators [80,81]. Likewise, mitochondrial injury and impaired OXPHOS might also alter METs [82], and the reduced OXPHOS in cGAS^−/−^ macrophages compared with WT might cause the lower METs in cGAS^−/−^ cells. Nevertheless, the study on METosis (cell death from METs) in sepsis is still lacking, and the data on the role of METs in the clinical setting of sepsis are still too limited. Although it is still unclear if METosis improves or worsens the conditions of sepsis [80], the different levels of METosis between WT and cGAS^−/−^ macrophages after LPS induction imply the involvement of the cGAS pathway in METs. Notably, both CitH3 and METosis are the general inflammatory responses resulting from the downstream signaling of several pathways (e.g., cGAS, TLR, and inflammasome), and the different levels of both parameters between cGAS^−/−^ and WT mice imply that cGAS is one of these pathways inducing CitH3 and METosis. Because the inducers of cGAS receptors are cytosolic DNA, both mtDNA and host DNA can activate cGAS [83]. With more prominent LPS-induced mitochondrial stress in WT macrophages than cGAS^−/−^ cells (Figure 7), mtDNA may be the important cGAS inducer supporting previous publications [84]. Future experiments with DNase treatment and/or exogenous cGAMP rescue might provide a more definitive conclusion rather than the current suggestive assumption.

In conclusion, we demonstrated proof of concept that damaged DNA in sepsis might enhance systemic inflammation through the cGAS receptor. Although cGAS is a sensor of cytosolic DNA, the deletion of cytosolic DNA by cGAS in macrophages induced several alterations compared with the WT cells, especially after LPS stimulation. The anti-inflammation of cGAS^−/−^ macrophages compared with WT implies the possible use of cGAS inhibitors in sepsis. More studies are warranted.

Several limitations must be mentioned. First, the systemic ablation of cGAS mice may also affect other innate and adaptive immune cells. The potential contribution of these cell types to the observed phenotypes has not been examined in this study. More studies on these topics would be valuable. Second, the use of cGAS^−/−^ mice, with a deficiency of the cGAS gene in all cells, not only in the macrophages, does not fully support the importance of the cGAS gene in the macrophages. Employing the LysM-Cre system to achieve targeted deletion of cGAS only in macrophages would provide a more direct approach to investigate the function of cGAS in catheter-associated inflammation. Third, the in vitro experiments were performed using macrophages from the bone marrow, but not the injured organs (liver and kidney). While bone marrow-derived macrophages represent systemic inflammatory responses, they might be different from macrophages isolated from the internal organs (the resident macrophages) [85]. Fourth, the use of a cGAS inhibitor was not performed, partly due to the limitation of the inhibitor, which shows less effective blockage of cGAS functions compared with cGAS^−/−^ mice. More experiments using effective cGAS inhibitors would be valuable.

## 4. Materials and Methods

### 4.1. Animals and Animal Model

The procedure and protocol for animal study were adapted from the United States National Institutes of Health (NIH) animal care and use protocol. It was approved by the Institutional Animal Care and Use Committee of the Faculty of Medicine, Chulalongkorn University (SST 025, 2563). Wild-type C57BL/6J (WT) mice were purchased from Nomura Siam (Pathumwan, Bangkok, Thailand), and cGAS knockout (cGAS^−/−^) mice in a C57BL/6J background were kindly provided by Professor Søren Riis Paludan (Aarhus University, Aarhus, Denmark). Only male 8-week-old mice, weighing approximately 20–22 g, housed in standard clear plastic cages (3–5 mice per cage) with free access to water and food (SmartHeart Rodent; Perfect Companion Pet Care, Bangkok, Thailand) in light/dark cycles of 12/12 h in 22 ± 2 °C, with 50 ± 10% relative humidity and thick paper stripes for environmental enrichment, were used.

The catheter-induced sepsis model using the *P. aeruginosa*-containing catheter, following our previous protocol, was performed as previously described [23,86]. Briefly, a clinical, isolated strain of *Pseudomonas aeruginosa* from a blood sample of patients (pneumonia-induced septicemia) in the King Chulalongkorn Memorial Hospital (Bangkok, Thailand) with informed consent was obtained, and the same strain was used in all the experiments. Notably, the isolated *P. aeruginosa* was confirmed by the regular procedure of the hospital using Gram stain, standard conventional biochemical tests, and the commercial kits, including VITEK^®^2 and VITEK^®^MS (bioMérieux, Marcy-l’Étoile, France). Then, the bacteria were grown in Tryptic Soy broth (TSB) (Oxoid Ltd., Basingstoke, Hampshire, England) for 24 h at 37 °C, resuspended in phosphate buffered saline (PBS; pH 7.4), adjusted to the turbidity of 0.5 McFarland standard per 1 mL PBS (approximately 1 × 10^8^ CFU/mL) in TSB, and incubated at 37 °C for 4 h on 6-well plates (5 mL of TSB/well) with 25 mm segments of polyurethane catheter (NIPRO, Ayutthaya, Thailand). Subsequently, the catheters were washed with PBS and subcutaneously implanted onto the mouse flank on both sides under isoflurane anesthesia. In the control group, the catheters were processed with the same procedure as the infected catheter without bacterial resuspension using sterile TSB and washed with PBS before subcutaneous insertion. At 5 days after catheter insertion, mice were sacrificed with sample collection by cardiac puncture under isoflurane anesthesia. Blood was mixed with 3% volume by volume (*v*/*v*) of acetic acid in a ratio of 1:20 for red blood cell lysis before counting with a hemocytometer, and the total number of these cells was calculated by the total count from a hemocytometer multiplied by the percentage of cells from the Wright-stained blood slide. Additionally, blood bacterial abundance (bacteremia) was determined by direct spreading mouse blood onto blood agar plates (Oxoid, Hampshire, UK) in serial dilutions and incubating at 37 °C for 24 h before colony enumeration. Serum was measured for serum creatinine (renal injury) and liver enzymes (aspartate transaminase and alanine transaminase) with QuantiChrom Creatinine (DICT-500), EnzyChrom Aspartate Transaminase (EALT-100), and EnzyChrom Alanine Transaminase (EALT-100) (Bioassay, Hayward, CA, USA). On the other hand, serum cytokines and endotoxin (LPS) were determined by enzyme-linked immunosorbent assay (ELISA) (Invitrogen, Carlsbad, CA, USA) and the Limulus amebocyte lysate test (Associates of Cape Cod, East Falmouth, MA, USA), respectively, while cell-free DNA (cfDNA) was measured by Quanti PicoGreen assay (Sigma-Aldrich).

For further support of the different sepsis severity between cGAS^−/−^ and WT mice, the lethal dose of lipopolysaccharides (LPS) was used [30]. As such, intraperitoneal (IP) injections of 25 mg/kg LPS from *Escherichia coli* 026:B6 (Sigma-Aldrich, St. Louis, MO, USA) were performed. Mice were sacrificed at 3 h post-LPS injection by cardiac puncture under isoflurane anesthesia, while some mice were monitored for 48 h for survival analysis.

### 4.2. Flow Cytometry Analysis

Flow cytometry analysis was performed according to the previously published protocol [87]. Single-cell suspension of 1 × 10^6^ splenocytes was stained with surface antibodies (Biolegend, San Diego, CA, USA), including anti-F4/80 (clone BM8; cat. 123110), anti-CD80 (clone 16-10A1; cat. 104706), anti-CD86 (clone PO3; cat. 105128), anti-CD163 (clone S15049l; cat. 155320), anti-IAb (clone AF6-120-1; cat. 116406), anti-CD11b (clone M1/70; cat. 101228), anti-Ly6g (clone 1A8; cat. 127608), and anti-Gr-1 (clone RB6-8C5; cat. 108412), for 20 min in the dark. In addition, Fixable Viability Dye eFlour^TM^ 780 (Invitrogen) was used for excluding dead cells. After fixation, antibody-labeled splenocytes were permeabilized for 10 min and incubated with anti-CD206 (clone C068C2; cat. 141708, Biolegend), which acted as an intracellular antibody, for 30 min in the dark. Isotype controls were used for each antibody to determine the gate. Then, 30,000 stained cells were acquired per sample and gated on live cells using a FACS LSR II cytometer (BD Biosciences, Franklin Lakes, NJ, USA) with FlowJo version 10 (Ashland, Wilmington, DE, USA). The flow cytometry results were demonstrated in the percentage of the interested cells, and the absolute cell count could be calculated using the percentage of cells multiplied by 30,000.

### 4.3. The Transcriptomic Analysis

Bone marrow-derived macrophages (BMDMs) from WT and cGAS^−/−^ mice were generated [88] before the in vitro activation as previously described [18]. Briefly, LPS (150 ng/mL of *E. coli* 026:B6 LPS) (Sigma-Aldrich, St. Louis, MO, USA) or media control was added for 24 h before the analysis of supernatant and cells, as previously published [12]. For transcriptome analysis, the macrophage RNA was extracted by the RNeasy Mini Kit (Qiagen, Hilden, Germany) and processed with the RNA library preparation and sequencing using Illumina NextSeq 500 at the Omics Sciences and Bioinformatics Center, Chulalongkorn University, Bangkok, Thailand [2]. The raw sequencing reads were mapped against the *Mus musculus* reference genome GRCm39. Reads were mapped and aligned with STAR [89]. Then, they were counted by Kallisto quantification [90]. All the sequence quality, mapping, alignment, and quantification were performed on the Galaxy platform [91]. After normalization, differentially expressed genes (DEGs) were retrieved using the edgeR [92] and limma [93] packages in R version 4.0.4. Analyses were conducted on triplicate samples. Genes were considered differentially expressed when the log2 fold-change was less than −2 or greater than 2 (representing down- or upregulation, respectively) and the BH-adjusted *p*-value was less than 0.05. The selected DEGs were clustered based on Euclidean distance and the Ward.D2 method. Thereafter, up-regulated DEGs selected from each macrophage group were analyzed for their function using gene set enrichment analysis (GSEA) [94] and gene sets from the MSigDB database [95]. The pathway activity was predicted using PROGENy and DoRothEA based on the transcriptomic profile [35,36]. The macrophage groups were classified into different macrophage subgroups using the bioinformatic tool MacSpectrum [42]. The RNA-seq data set is available at the GEO database (GSE284706).

### 4.4. The In Vitro Analysis

Supernatant cytokines (TNF-α, IL-6, and IL-10) and cyclic guanosine monophosphate–adenosine monophosphate (2′3′-cGAMP) were determined by ELISA (Invitrogen) and an assay from Cayman Chemical (Ann Arbor, MI, USA), respectively. Real-time quantitative reverse transcription polymerase chain reaction (qRT–PCR) was performed following a previous protocol using the total RNA prepared by Trizol, quantified by NanoDrop ND-1000 (Thermo Fisher Scientific, Waltham, MA USA), and converted into cDNA by Reverse Transcription System in SYBR Green system (Applied Biosystem, Foster City, CA, USA). The cDNA templates and target primers were analyzed by the ∆∆CT method (2^−∆∆Ct^) relative to the *β-actin* housekeeping gene. In parallel, mitochondrial enumeration was evaluated by mitochondrial DNA (mtDNA) using a tissue genomic DNA extraction mini kit (Favorgen Biotech, Wembley, WA, Australia) and NanoDrop ND-100 (Thermo Fisher Scientific), normalized by *β2-microglobulin* (*β2M*) as previously described [96]. Also, mitochondrial biogenesis was analyzed by MitoTracker assay with 200 nM of MitoTracker Red CMxROS (Molecular Probes, Inc., Eugene, OR, USA), which was incubated at 37 °C for 15 min before fixing with cold methanol at −20 °C and measured by microplate reader at excitation OD579 nm and emission OD599 nm, as previously described [96]. All of the primers used are listed in Table 1.

Additionally, extracellular flux analysis using Seahorse XFp Analyzers (Agilent, Santa Clara, CA, USA) was performed for the cell energy analysis [2,3,4]. In brief, the stimulated macrophages at 1 × 10^5^ cells/well were incubated in Seahorse media (DMEM complemented with glucose, pyruvate, and L-glutamine) (Agilent, 103575-100) for 1 h before activation by different metabolic interference compounds, including oligomycin, carbonyl cyanide-4-(trifluoromethoxy)-phenylhydrazone (FCCP), and rotenone/antimycin A, for OCR evaluation. In parallel, glycolysis stress tests were performed using glucose, oligomycin, and 2-Deoxy-d-glucose (2-DG) for ECAR measurement. The data were analyzed using Seahorse Wave 2.6 software based on the following equations: maximal respiration = OCR between FCCP and rotenone/antimycin A OCR after rotenone/antimycin A; and maximal glycolysis (glycolysis capacity) = ECAR between oligomycin and 2-DG ECAR after 2-DG. Moreover, macrophage extracellular traps (METs or METosis) were determined by supernatant citrullinated histone 3 (CitH3) using ELISA assay (Cayman Chemical) and immunofluorescent staining according to a previously published protocol [97]. Briefly, BMDMs (2.5 × 10^5^ cells) were placed on a coverslip, then stimulated with *E. coli* 026:B6 LPS (Sigma-Aldrich) or media before fixing with 4% paraformaldehyde in 1xPBS and blocking for 30 min. with 1xTBS with 2% bovine serum albumin. The extracellular trap (ET) formation was detected with rabbit anti-histone H3 in dilution 1:300 (Abcam, Cambridge, United Kingdom) with secondary antibody (goat anti-rabbit IgG conjugated Alexa Fluor 488) (Abcam Cambridge, United Kingdom), while DAPI (1 µg/mL) was used to detect the DNA and visualized by a fluorescence microscope (Olympus, Tokyo, Japan).

### 4.5. Statistical Analysis

The normality test to support the parametric characteristic of the data was performed by the Kolmogorov–Smirnov test (*p* value less than 0.05). Mean ± standard error of the mean (SEM) was presented using the one-way analysis of variance (ANOVA) followed by Tukey’s analysis for multiple group comparison or the Wilcoxon test for pairwise comparison. Analysis of the time-point data was determined by the repeated measures ANOVA. Survival analysis was performed using the log-rank test. All statistical analyses were performed using GraphPad Prism version 10.0 software (La Jolla, CA, USA), and a *p*-value of <0.05 was considered statistically significant.

## 5. Conclusions

In conclusion, cGAS^−/−^ mice with catheter-induced infection demonstrated similar burdens of infection (bacteremia and endotoxemia) but, surprisingly, less severe sepsis than WT (serum cytokines and liver enzymes). The transcriptomic analysis clearly indicated different responses against LPS between cGAS^−/−^ and WT macrophages, especially the upregulated genes for OXPHOS and mTORC1, which might lead to anti-inflammatory macrophages. The blockage of cGAS and/or macrophage manipulation may represent promising strategies for sepsis treatment in certain contexts. Further studies are warranted.

## Figures and Tables

**Figure 1 ijms-26-05069-f001:**
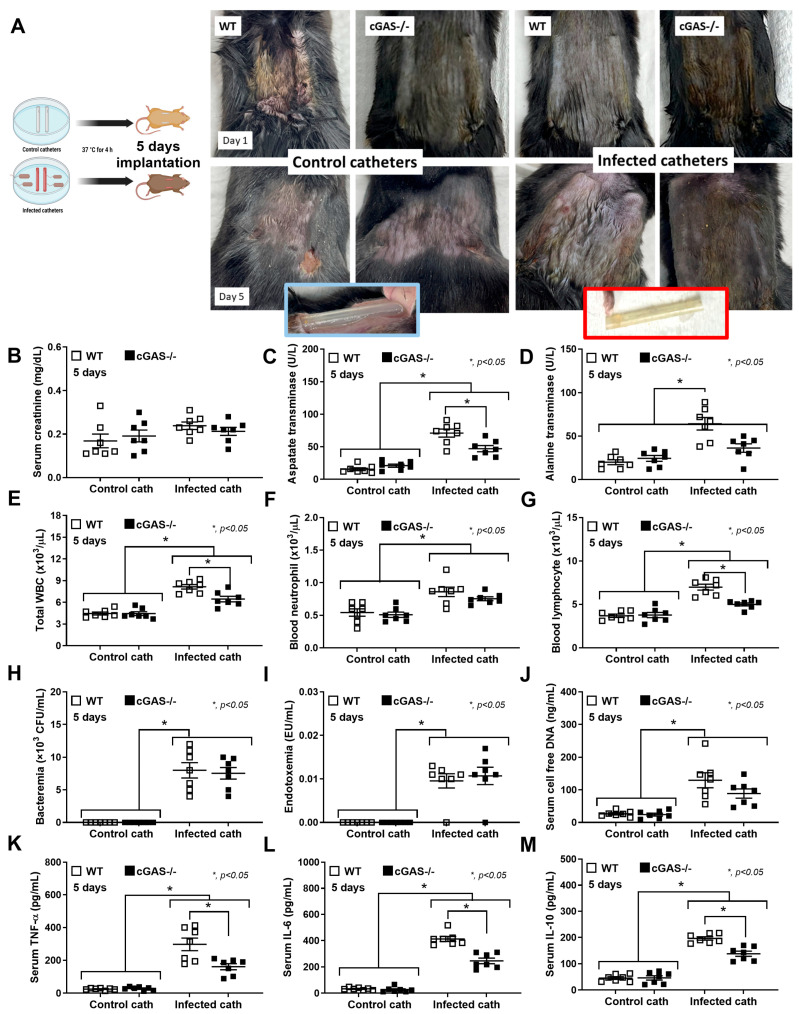
Severity of *P. aeruginosa*-infected catheter-induced sepsis model in WT and cGAS^−/−^ mice. (**A**) Schematic diagram of the sepsis model (left side) and the representative characteristics of the model (the pictures in the blue and red frames represent the clearer control catheters and the blue-green pus color of the infected catheters at 5 days of the models, respectively) (right side). Disease severity, as evaluated by (**B**) kidney injury (serum creatinine), (**C**,**D**) liver damage (serum aspartate transaminase and serum alanine transaminase), (**E**) total white blood cells (WBC), (**F**,**G**) blood neutrophil and lymphocyte count, (**H**) bacteremia, (**I**) endotoxemia, (**J**) cell-free DNA, and (**K**–**M**) serum cytokine levels (TNF-α, IL-6, and IL-10), is demonstrated (*n* = 7 per group as biological replication). The differences between groups were examined by one-way ANOVA followed by Tukey’s analysis; *, *p* < 0.05 indicates statistical significance between the paired groups. The cytokine levels are shown in Appendix A (Upper table).

**Figure 2 ijms-26-05069-f002:**
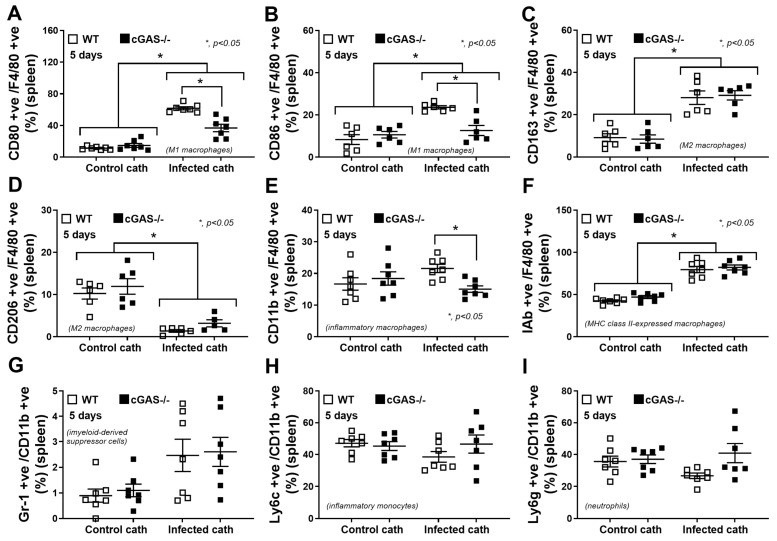
Flow cytometry analysis of splenocytes from WT and cGAS^−/−^ mice in the *P. aeruginosa*-infected catheter-induced sepsis model. (**A**,**B**) Macrophages were identified as F4/80-positive cells with M1 polarization (CD80 and CD86), (**C**,**D**) M2 polarization (CD163 and CD206), (**E**) inflammatory macrophage (CD11b), (**F**) MHC class II expression (IAb). Additionally, (**G**) myeloid-derived suppressor cells (Gr-1 and CD11b positive), (**H**) inflammatory monocytes (Ly6c and CD11b positive), and (**I**) neutrophils (Ly6g and CD11b positive) were analyzed (*n* = 7 per group as biological replication). The differences between groups were examined by one-way ANOVA followed by Tukey’s analysis; *, *p* < 0.05 indicates statistical significance between the paired groups. The gating strategy is shown in Appendix A.

**Figure 3 ijms-26-05069-f003:**
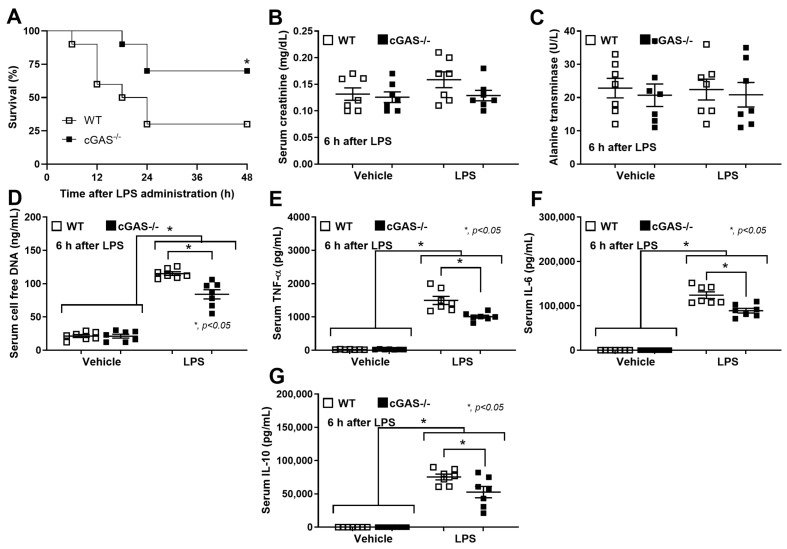
Severity of the LPS-induced sepsis model in WT and cGAS^−/−^ mice, as evaluated by (**A**) mortality, (**B**) kidney injury (serum creatinine), (**C**) liver damage (serum aspartate transaminase), (**D**) cell-free DNA, and (**E**–**G**) serum cytokine levels (TNF-α, IL-6, and IL-10) are demonstrated (*n* = 10 per group for survival analysis and *n* = 7 per group for other parameters as biological replication). The differences between groups were examined by one-way ANOVA followed by Tukey’s analysis; *, *p* < 0.05 indicates statistical significance between the paired groups. The cytokine levels are shown in Appendix A (Lower table).

**Figure 4 ijms-26-05069-f004:**
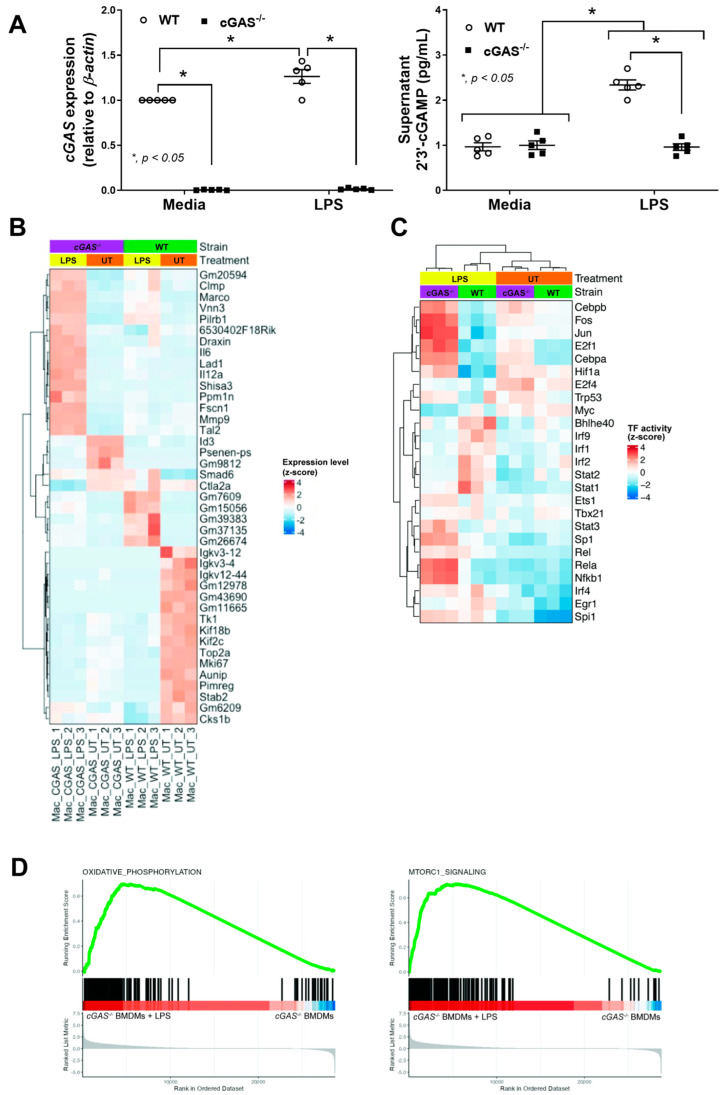
Transcriptomic profiling of wild-type (WT) and cGAS^−/−^ macrophages in the presence of lipopolysaccharide (LPS). (**A**) Gene expression level of *cGAS* and amount of secreted 2′3′-cGAMP, (**B**) heatmap of normalized gene expression level (z-score), (**C**) the predicted transcription factor (TF) activity based on gene expression profile, and (**D**) waterfall plots of selected pathways from gene set enrichment analysis (GSEA) are demonstrated (*n* = 3 per group as technical replicates). The differences between groups were examined by one-way ANOVA followed by Tukey’s analysis; *, *p* < 0.05 indicates statistical significance between the paired groups.

**Figure 5 ijms-26-05069-f005:**
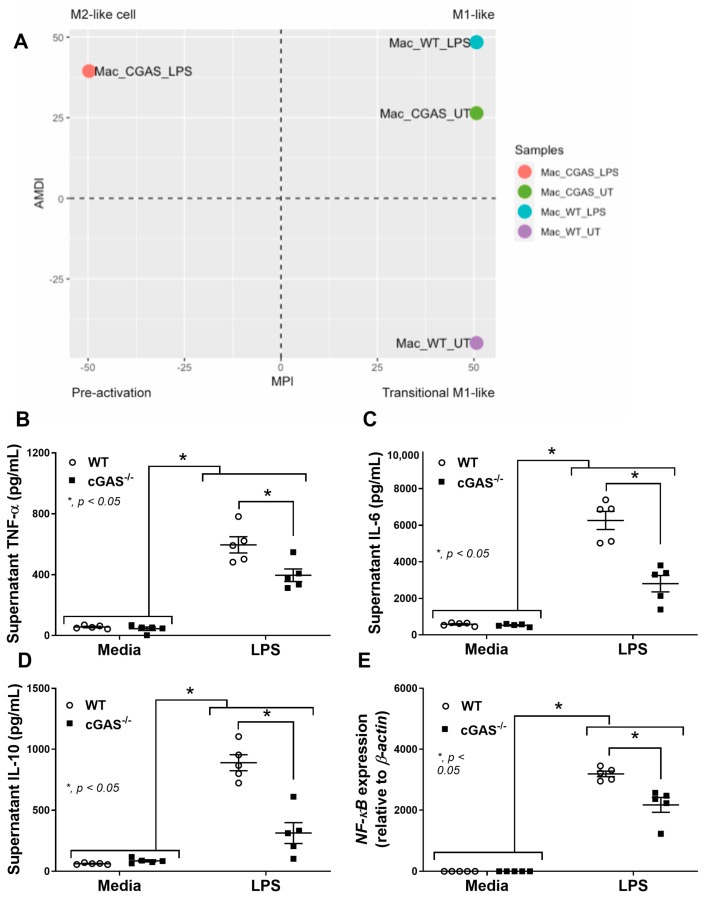
cGAS deletion leads to more anti-inflammatory characteristics of cGAS^−/−^ macrophages than wild-type (WT) in the presence of lipopolysaccharide (LPS). (**A**) Predicted macrophage subtype by bioinformatic tool (the MacSpectrum) based on gene expression profile of WT and cGAS^−/−^ BMDMs after LPS stimulation, (**B**–**D**) supernatant cytokines (TNF-α, IL-6, and IL-10), and (**E**) the expression of nuclear factor kappa B (*NF-κB*) transcriptional factor are demonstrated. The results are derived from the independently isolated experiments. The differences between groups were examined by one-way ANOVA followed by Tukey’s analysis; *, *p* < 0.05 indicates statistical significance between the paired groups.

**Figure 6 ijms-26-05069-f006:**
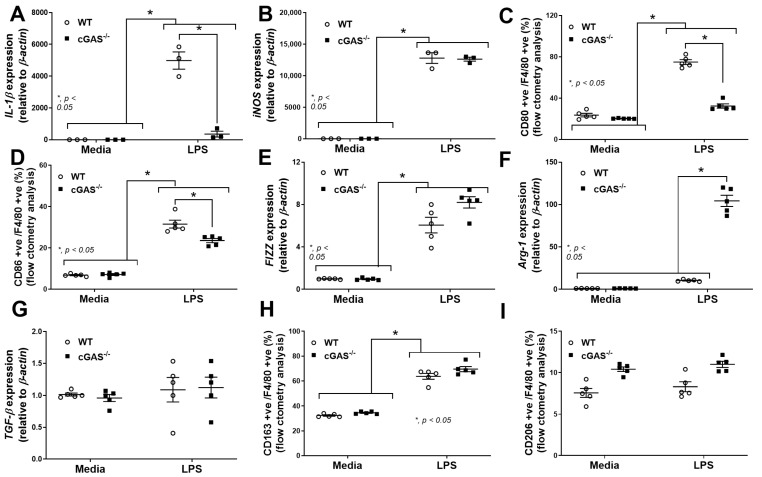
cGAS deletion leads to the M2-like phenotype in cGAS^−/−^ macrophages compared to wild-type (WT) in the presence of lipopolysaccharide (LPS). (**A**–**D**) The M1 macrophage polarization parameters, including gene expression of *IL-1β* and *iNOS* (using PCR) with the flow cytometry markers (CD80 and CD86 positive with F4/80), and (**E**–**I**) M2 macrophage polarization parameters, including *Fizz*, *Arg-1*, and *TGF-β*, with the flow cytometry markers (CD163 and CD206 positive with F4/80). The results are derived from the independently isolated experiments (technical replication). The differences between groups were examined by one-way ANOVA followed by Tukey’s analysis; *, *p* < 0.05 indicates statistical significance between the paired groups. The gating strategy is shown in Appendix A.

**Figure 7 ijms-26-05069-f007:**
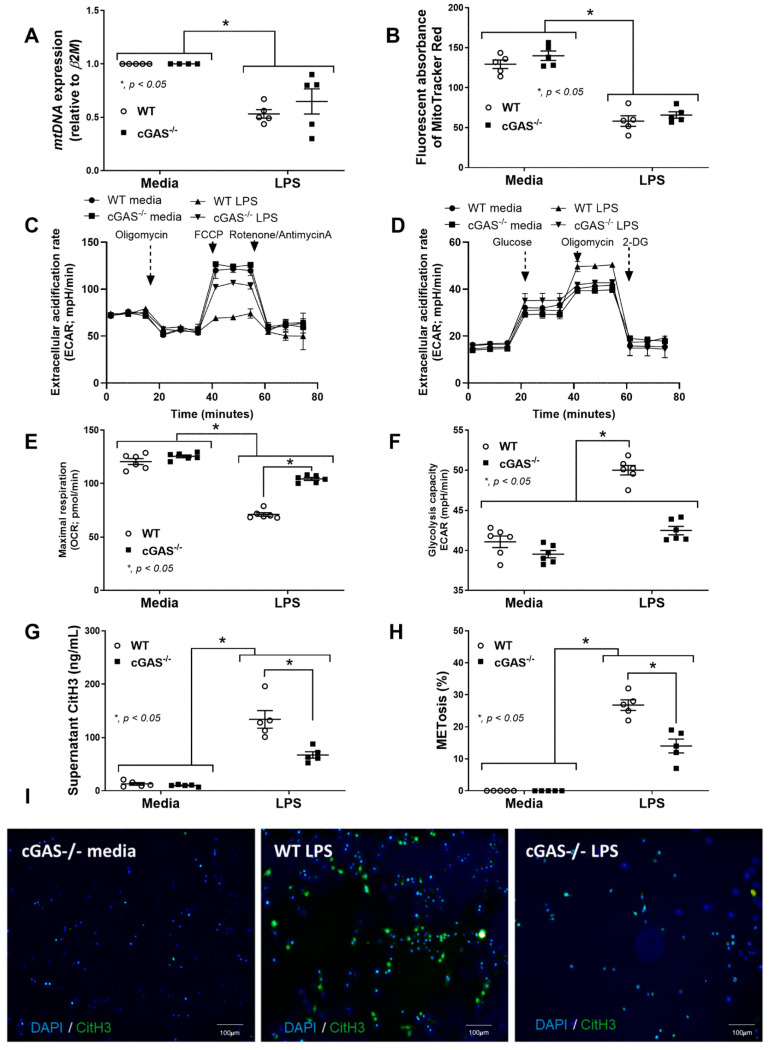
cGAS deletion leads to similar mitochondrial DNA (mtDNA) damage but lower macrophage extracellular traps (METs) in cGAS^−/−^ macrophages compared to wild-type (WT) in the presence of lipopolysaccharide (LPS). (**A**) The mitochondrial DNA (mtDNA), (**B**) mitochondrial damage (MitoTracker red), (**C**) oxygen consumption rate (OCR) (as determined by a mitochondrial stress test), (**D**) extracellular acidification rate (ECAR) (determined by a glycolysis stress test), (**E**,**F**) graphical presentation of maximal respiration and maximal glycolysis, and (**G**) METosis parameters, using supernatant citrullinated histone 3 (CitH3), as well as (**H**,**I**) percentage of the fluorescent stained CitH3-positive cells (green colors) with the representative pictures, are demonstrated. The results are derived from the independently isolated experiments (technical replication). The differences between groups were examined by one-way ANOVA followed by Tukey’s analysis; *, *p* < 0.05 indicates statistical significance between the paired groups; the METs of WT with media are not shown due to the same negative CitH3 staining between WT and cGAS^−/−^ with media control; blue color, DNA stained using 4′,6-diamidino-2-phenylindole (DAPI).

**Table 1 ijms-26-05069-t001:** List of primers in the study.

Primers		Sequences	Accession Number
Cyclic GMP–AMP synthase (*cGAS*)	Forward	5′-ATGTGAAGATTTCGCTCCTAATGA-3′	KC294567.1
	Reverse	5′-GAAATGACTCAGCGGATTTCCT-3′	
Nuclear factor kappa B (*NF-κB*)	Forward	5′-CTTCCTCAGCCATGGTACCTCT-3′	M61909.1
	Reverse	5′-CAAGTCTTCATCAGCATCAAACTG-3′	
Interleukin-1β (*IL-1β*)	Forward	5′-GAAATGCCACCTTTTGACAGTG-3′	NM_008361.4
	Reverse	5′-TGGATGCTCTCATCAGGACAG-3′	
Inducible nitric oxide synthase (*iNOS*)	Forward	5′-ACCCACATCTGGCAGAATGAG-3′	AF427516.1
	Reverse	5′-AGCCATGACCTTTCGCATTAG-3′	
Resistin-like molecule-α (*Fizz-1*)	Forward	5′-GCCAGGTCCTGGAACCTTTC- 3′	AF323082.1
	Reverse	5′-GGAGCAGGGAGATGCAGATGA-3′	
Arginase-1 (*Arg-1*)	Forward	5′-CTTGGCTTGCTTCGGAACTC-3′	NM_007482.3
	Reverse	5′-GGAGAAGGCGTTTGCTTAGTTC-3′	
Transforming growth factor (*TGF-β*)	Forward	5′-CAGAGCTGCGCTTGCAGAG-3′	AH003562.3
	Reverse	5′-GTCAGCAGCCGGTTACCAAG-3′	
Beta-actin *(β-actin)*	Forward	5′-CGGTTCCGATGCCCTGAGGCTCTT-3′	NM_007393.5
	Reverse	5′-CGTCACACTTCATGATGGAATTGA-3′	
Mitochondrial DNA (*mtDNA*)	Forward	5′-CGTACACCCTCTAACCTAGAGAAGG-3′	PV231059.1
	Reverse	5′-GGTTTTAAGTCTTACGCAATTTCC-3′	
β2-microglobulin (*β2M*)	Forward	5′-TTCTGGTGCTTGTCTCACTGA-3′	NM_009735.3
	Reverse	5′-CAGTATGTTCGGCTTCCCATTC-3′	

## Data Availability

The data are contained within the article.

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
