# Peer review of "Cyclic GMP-AMP Synthase (cGAS) Deletion Promotes Less Prominent Inflammatory Macrophages and Sepsis Severity in Catheter-Induced Infection and LPS Injection Models"

_ijms, 2025, doi:10.3390/ijms26115069_

Round 1

Reviewer 1 Report

Comments and Suggestions for Authors

Suksamai et al. present a new immunological study investigating the role of cGAS as a promoter of sepsis severity in mice, they used animal models together with in vitro approaches to investigate their hypothesis. They performed additional FACS and transcriptomic profiling of macrophages revealing the activation status. I went through this manuscript and provide the following comments:

Major Points:

  1. Although the authors investigate the sepsis, they used a wrong animal model. Using the subcutaneous catheter infection is not appropriate for the study. This makes the study irrelevant for the public health, since there are no clinical conditions where a catheter would be inserted subcutaneously in patients. To test this, the authors need to use an intravenous catheter for example.
  2. Beside using the wrong model, the authors used the wrong conditions and the wrong medium. Incubating the catheter with the bacterium for 4 hrs before inserting them in the mice does not fit with the study. It would be logical to insert the catheter and then inject the inoculant. In addition, using TSB is wrong. This medium is not physiological and contain a high percentage of glucose which will allow the bacteria to aggregate together forming biofilms. This cannot clearly indicate whether the reactions mentioned later are because of the bacteria initiating the infection or because of the biofilm-forming bacteria which are ready to react immunologically. The authors should also provide pictures of the flank regions of their control and infected groups, in order to be able to judge the local effect of inserting sterile/contaminated catheter. Even inserting a sterile catheter in the flank region should induce some immunological reactions because of the skin microbiota e.g. aureus which will proliferate upon inducing a breach in the skin.
  3. The authors need to describe the clinical isolate used in the study. How was this isolate discovered and how was the isolation confirmed?
  4. Although the authors described the severity of the infection progress by analysing kidney- and liver-related parameters, they decided to work later with bone marrow derived macrophages! It would be meaningful to use renal and hepatic macrophages for the analysis later.
  5. The last author „Asada Leelahavanichkul” self-cited 25 papers of his own work in this manuscript! This is around 28.1% of the total references used in this Manuscript (25 out of 89). Some of the references are not necessary and taking about the COVID-19! The authors SHOULD remove all unnecessary references.

Minor Points:

  1. The abstract should be modified. The authors should avoid using the personal language. For example, using the words “might be” in line 20 does not seem interesting. In addition, it makes no sense to add the words “flow cytometry analysis” in line 25 or mentioning that further studies are interesting as in lines 37-38.
  2. The image in the supplementary information should be labelled and fully described.
  3. The authors need to provide a description of the statistical test used in every figure. It would be for the reader to understand the figures and judge them.
  4. In line 50, please provide a description of the abbreviation “GMP-AMP”.
  5. The authors need to indicate the gating strategy of their FACS analyses.
  6. Why did the authors choose to show the percentage of cells in figure 2? Were the absolute cell counts also similar to the showed percentages?
  7. If there is no difference in the bacteraemia in figure 1H, how did the authors build their assumption that deletion of cGAS reduce the sepsis severity?
  8. I somehow confused. The authors used LPS in some experiments to prove their hypothesis, although they used individual cells of aeuroginosa in their infection model! Using LPS is completely different than using the bacteria themselves (even though LPS is a part of the bacteria).

Reviewer 2 Report

Comments and Suggestions for Authors

The authors used cAGS knockout mice to demonstrate that cGAS deficiency could reduce organ injury and inflammatory response in mice with a catheter-induced infection. The idea is good. However, the mouse model of sepsis is not a conventional one. Most accepted model is still CLP model for sepsis, or LPS model for endotoxemia. It would be better to include these two models to have a solid data for publication.

In addition, cGAS is the receptors for DNAs particularly for host genomic DNA or mtDNA. The most likely explanation is the deficiency of cGAS that blocks the DNA-cGAS-STING pathway to reduce the inflammation and organ injury, instead of the transition of macrophages.

The FACS graphs are not standard. The values of cytokines shall be given in a table.

Comments on the Quality of English Language

The manuscript shall be corrected by native English speaker.

Reviewer 3 Report

Comments and Suggestions for Authors

This manuscript studies the role of cGAS in modulating macrophage inflammatory responses during catheter-induced sepsis and LPS stimulation. The authors employ a combination of in vivo infection modeling and transcriptomic profiling to argue that cGAS deficiency leads to reduced inflammation and a shift toward an M2-like macrophage phenotype.

The overall experimental framework is sound, and the question is biologically relevant. However, several key interpretations are either internally inconsistent, not supported by the data, or insufficiently validated. Below I outline the major concerns:

  1. The authors report upregulation of Il6 and Il12 transcripts and increased NF-κB and MAPK transcription factor activity in cGAS-deficient macrophages—features typically associated with M1 polarization. Although IL-6 protein levels were lower, the explanation of post-transcriptional regulation remains speculative. Likewise, the suggestion that NF-κB may mediate anti-inflammatory functions in this context is not well supported given the extensive literature showing its pro-inflammatory role in LPS-treated macrophages.
  2. The conclusion that cGAS deficiency skews macrophages toward an M2-like state relies heavily on MacSpectrum output and elevated Arg1 However, other widely accepted M2 markers (CD206TGF-βIL-10) are either unchanged or not shown. These markers are routinely used in similar LPS-based settings across the literature. Without broader validation, the claim of M2 polarization is premature.
  3. The manuscript includes statements such as “cGAS-deficient macrophages respond more strongly to DNA” and “cGAS protects against mtDNA damage,” which are biologically confusing. cGAS is a DNA sensor, so its absence would be expected to reduce, not enhance, DNA responsiveness. This should be clarified.
  4. The authors suggest that mTORC1 enrichment is associated with M2 polarization in cGAS-deficient macrophages. However, previous studies show that cGAS activation promotes mTORC1 signaling and M1-like responses. Without direct metabolic data, this interpretation is speculative.
  5. While CitH3 is a recognized marker of extracellular trap formation, it reflects a general inflammatory response and is not specific to mtDNA damage or cGAS signaling. The authors should be more cautious in linking METosis directly to mtDNA sensing in the absence of functional validation.
  6. The authors made a conclusion that increased inflammation in WT cells is due to mtDNA sensing by cGAS. However, LPS activates many pathways beyond cGAS (e.g., TLR4, inflammasome). Additionally, mtDNA abundance (by qPCR) and mitochondrial damage (MitoTracker) appear comparable between groups. Without DNase treatment, exogenous cGAMP rescue, or other functional tests, this conclusion remains suggestive rather than definitive.
  7. Please provide specific figure references for each measurement in text. For instance, the sentence describing neutrophil abundance, bacteremia, endotoxemia, and cfDNA (Fig. 1B–M) is difficult to follow because these data are spread across several panels that are not clearly linked in the text.
  8. The flow cytometry plots in Fig. 2J are very low resolution and lack percentage values. Please provide higher-resolution images and annotate population percentages.

In summary, the manuscript addresses an important topic and provides useful data on macrophage responses in sepsis. However, several central conclusions—particularly regarding macrophage polarization, cGAS-mTORC1 signaling, and DNA sensing—require more cautious interpretation and additional experimental support.

Reviewer 4 Report

Comments and Suggestions for Authors

Dear Authors,

Thank you for the opportunity to review your manuscript investigating the role of cyclic GMP-AMP synthase (cGAS) in modulating immune responses during sepsis induced by Pseudomonas aeruginosa infection in a catheter-based murine model. Your study combines transcriptomic, immunologic, and cellular analyses to provide new insights into how cGAS deficiency attenuates inflammatory macrophage activation, dampens cytokine responses, and reduces macrophage extracellular trap (MET) formation. The work is timely, relevant, and addresses an important question in immunometabolism and sepsis pathophysiology.

In the attached review, I provide detailed comments to enhance the clarity, rigor, and impact of your manuscript.

Round 2

Reviewer 2 Report

Comments and Suggestions for Authors

The authors have answered most of the important comments

Reviewer 3 Report

Comments and Suggestions for Authors

The authors have addressed the majority of my concerns. I recommend the manuscript for publication.